# Translation and Cross-Cultural Adaptation of the Pediatric Outcomes Data Collection Instrument into the Italian Language

**DOI:** 10.3390/children9060853

**Published:** 2022-06-08

**Authors:** Giovanni Trisolino, Stefano Stallone, Paola Zarantonello, Andrea Evangelista, Manila Boarini, Jacopo Faranda Cordella, Luca Lerma, Luisa Veronesi, Cosma Caterina Guerra, Luca Sangiorgi, Giovanni Luigi Di Gennaro, Renato Maria Toniolo

**Affiliations:** 1Pediatrics Orthopedics and Traumatology, IRCCS Istituto Ortopedico Rizzoli, 40136 Bologna, Italy; giovanni.trisolino@ior.it (G.T.); stallone.stefano@gmail.com (S.S.); p.zarantonello1@gmail.com (P.Z.); luca.lerma@ior.it (L.L.); luisa.veronesi@ior.it (L.V.); cosmacaterina.guerra@ior.it (C.C.G.); giovanniluigi.digennaro@ior.it (G.L.D.G.); 2Unit of Clinical Epidemiology, CPO Piemonte, AOU Città della Salute e della Scienza di Torino, 10126 Turin, Italy; andrea.evangelista@cpo.it; 3Department of Rare Skeletal Disorders, IRCCS Istituto Ortopedico Rizzoli, 40136 Bologna, Italy; luca.sangiorgi@ior.it; 4Clinical Trial Centre, IRCCS Istituto Ortopedico Rizzoli, 40136 Bologna, Italy; jacopo.farandacordella@ior.it; 5Department of Orthopedics and Traumatology, IRCCS Ospedale Pediatrico Bambino Gesù, 00146 Rome, Italy; renatomaria.toniolo@opbg.net

**Keywords:** PODCI, HSS Pedi-FABS, outcomes, quality of life, cross-cultural adaptation, musculoskeletal disorders, child, patient reported, parent-reported, Italian

## Abstract

(1) Background: The Pediatric Outcomes Data Collection Instrument (PODCI) is an English-language questionnaire specifically designed to assess health-related quality of life in children and adolescents with musculoskeletal disorders. This scoring system has been translated into several languages. Given the lack of an Italian version of the PODCI, this study aimed to translate, cross-culturally adapt, and assess the psychometric properties of the PODCI score in the Italian pediatric population. (2) Methods: The PODCI questionnaire was culturally adapted to Italian patients in accordance with the literature guidelines. The study included 59 participants from a single orthopedic institution who underwent orthopedic surgery for various skeletal conditions. The questionnaire was administered to participants at multiple time-points (T0, T1, T2). Internal consistency was evaluated using Cronbach’s alpha. Reproducibility was assessed using the intraclass correlation coefficient (ICC) between T0 and T1 assessment. Criterion validity was assessed using Spearman’s correlation coefficients between PODCI and the Hospital for Special Surgery Pediatric Functional Activity Brief Scale (HSS Pedi-FABS). Responsiveness was evaluated by the difference between T0 and T2 using the effect size (ES) and the standardized response mean (SRM) calculation. (3) Results: Cronbach’s alpha was acceptable in both the self- and parent-reported versions with values of 0.78 (0.68–0.90) and 0.84 (0.60–0.92), respectively. The ICC fluctuated between 0.31 and 0.89 for self-reported and 0.49 to 0.87 for pediatrics. The Spearman’s r showed a moderate correlation between HSS Pedi-FABS and the “Sport & Physical Functioning” and “Global Functioning” domains. ES and SRM varied from small to moderate across all the domains. (4) Conclusions: This study demonstrates that the Italian version of the PODCI, translated following the international standardized guidelines, is reliable, valid, and responsive in pediatric patients who underwent orthopedic surgery.

## 1. Introduction

Medical care is focused to improve the patients’ condition in terms of satisfaction and wellness for themselves and their relatives. In order to obtain this purpose, the assessment of disease impact and the effect of treatments on patients’ well-being is pivotal, especially in children and adolescents [1]. From this perspective, the increasing use of questionnaires investigating both patients’ and parents’ perception enable an in-deep understanding of the disease, addressing the related therapeutic interventions [2].

Patient-Reported Outcomes Measurements (PROMs) are widely used in medicine. They are self- and/or proxy-reported tools that aim to measure the patient’s perspective on symptoms, pain, daily functions, psychological well-being, quality of life and social interaction of some categories of individuals (adults, children, adolescents, pregnant women, elderly population, categories of workers, athletes). They are useful for evaluating the impact of a disease, condition, treatment, and care program, having demonstrated high reliability, validity, and sensitivity to symptom change [3].

Many PROMs for adults have been developed and translated into several languages, including Italian. However, there is currently no formal translated questionnaire into Italian that addresses pediatric patients with orthopedic illnesses on a multidimensional level. This is a critical point for patients with several related disorders for whom unidimensional evaluation may be insufficient for a holistic health assessment.

Therefore, the RAMS (Rete Apparato Muscolo-Scheletrico), an Italian network comprised of the main leading national research institutions in pediatric orthopedics, launched an initiative to promote the use of standardized patient-reported measures that could be implemented on digital platforms and electronic devices, to share data across the network’s hospitals.

The “Pediatric Outcomes Data Collection Instrument” (PODCI) has been validated for over 20 years as a tool used to measure functional outcomes in pediatric orthopedics (aged 2 to 18 years) for a wide range of ages and conditions [4]. Moreover, it appeared to be well-suited for orthopedic surgeons to monitor their patients’ health and treatment efficacy at baseline and follow-up.

The PODCI is made up of three different formats: filled by parents for children, filled by parents for adolescents, and self-report for adolescents. The questionnaire is multidimensional, consisting of 86 questions (83 questions in adolescent self-report) that investigate the following domains: ‘upper extremity and physical function’ (UE), ‘transfer and basic mobility’ (TBM), ‘sports and physical function’ (SPF), ‘pain/comfort’ (PC), ‘happiness’ (H), ‘global functioning’ (GF) and ’expectations from treatment and satisfaction’ (E). The mean score for each domain is reported on a 0 (worst result) to 100 (best result) scale [4].

The PODCI has been used to investigate different orthopedic conditions and has proven to be extremely versatile and reliable [4,5,6,7,8,9,10,11,12,13,14,15]. It has already been validated in several languages [16,17,18,19,20,21].

To date, an official translation into Italian is still lacking. Therefore, the main purpose of the present study was to provide an official translation and cross-cultural adaptation in Italian of the PODCI. Concurrently, we assessed the psychometric properties of the PODCI in an Italian pediatric cohort of patients with different orthopedic conditions.

## 2. Materials and Methods

### 2.1. Translation and Cross-Cultural Adaptation

According to the American Academy of Orthopedic Surgeons (AAOS) and the international guidelines [22,23], the original English versions of the questionnaire were initially translated into Italian by two researchers, who are fluent in English and have knowledge about pediatric orthopedics (SS and PZ). Two independent senior expert researchers double-checked the draft for incongruences between the Italian and English versions; these researchers are fluent in English and experts in the field of orthopedic surveys (GT and GLDG). The PODCI questionnaires were back-translated from Italian to English by a native English translator who was uninformed of the study and the original questionnaire English versions. We obtained a new English version from the native translator, which we compared to the original to define a third correct Italian version: this step is pivotal in determining whether there was any eventual change or shift of significance related to linguistic expression during the translation process, as well as reconciling the differences. In accordance with the PODCI original version [4], we have considered that the Pediatric (2–10 years) and Adolescent parent-reported (11–18 years) questionnaires were overlapping in terms of the number of items and score for each domain opting to translate, adapt and validate the Pediatric version as well as the Adolescent self-report.

The final Italian PODCI formats, self- and parent-reported, were tested with a group of 15 patients and their parents with different pediatric orthopedic conditions, to check the appropriateness and clarity of the questionnaires and assess the cultural adaptation of the scale.

### 2.2. Study Design

The study was conceived as longitudinal prospective research. Ethics approval was sought and obtained from the local Ethical Committee (Study protocol: CE-AVEC 348/2021/Oss/IOR) in date 7 May 2021, and it was performed according to the Declaration of Helsinki and the guidelines for Good Clinical Practice. Parents were contacted by phone and given detailed information about the study; those who gave consent were included in the study.

#### 2.2.1. Participants

Between May 2021 and October 2021, 59 consecutively patients admitted to the Unit of Pediatric Orthopedics and Traumatology of a tertiary referral center for orthopedics were recruited.

According to the original PODCI version [4], male and female Italian mother-tongue patients aged 2 to 18 years with conditions ranging from high-frequency orthopedic functional impairments to chronic, progressive disorders who required surgical treatment were included in the study.

#### 2.2.2. Survey Design

Two surveys were developed for both self- and parent-reported questionnaires, including demographic and clinical characteristics (sex, age, diagnosis and planned surgical procedure) and PODCI, and they were administered in multiple time-points (T0 = baseline; T1 = 7-days follow-up, T2 = 3-months follow-up). Patients were requested to complete the surveys via e-mail, twice (T0 and T1) before surgery under stable clinical conditions, about 10 days apart, and a third time (T2) 3 months later. Furthermore, in order to obtain a valid and reliable anchoring tool for future validation of site or disease-specific questionnaires, patients were asked to complete the Italian version of the Hospital for Special Surgery Pediatric Functional Activity Brief Scale (HSS Pedi-FABS) [24], a sport activity rating scale validated for pediatrics [25], at baseline measurement (T0).

### 2.3. Psychometric Properties

The following properties were computed:

Internal consistency shows the homogeneity of the interrelatedness between the items; the responses to T0 were used to calculate the Cronbach’s alpha coefficient (α) (range 0–1). The higher the coefficient, the more consistent is the score, and values α ≥ 0.70 were deemed acceptable [26,27].

The consistency and accuracy of the evaluation when administered repeatedly under steady conditions are commonly used to assess reliability [28]. The overall scores at T0 and T1 were used to calculate the Standardized Error of Measurement (SEM) which indicates the measurement error at group level, the Smallest Detectable Change (SDC) which indicates the measurement error at individual level, and test–retest reliability with the interclass correlation coefficient r (ICC). An ICC ≥ 0.70 was considered acceptable [27].

The T0 scores were used to determine the criterion validity, which assesses the relationships between instruments. In the absence of a specified “gold standard” tool for the Italian pediatric population with musculoskeletal disorders, we anchored the subscales SPF and GF of the PODCI with the HSS Pedi-FABS, a previously tool validated in Italian that assesses the patient’s sport and general activity level. Correlation were measured using the Spearman’s correlation coefficient (ρ) (ρ < 0.3 low, 0.3 ≤ ρ ≤ 0.7 moderate, ρ > 0.7 excellent) [24].

Floor and ceiling effects occur when a score is inadequate to assess patients’ level of ability because of a huge number of lowest or highest scores; they were calculated from the aggregate T0 scores, and results were regarded acceptable if floor or ceiling effects were ≤30% [24].

The ability to detect clinically significant changes between the patients’ pre-intervention and post-intervention states is defined as responsiveness to change. It was calculated using a distribution-based method from the overall **T0** and **T2** scores of those patients who reported a minimal clinical improvement. The Standardized Response Mean (**SRM**) was calculated with the formula:(1)$$SRM=\frac{Mean\;T2-Mean\;T0}{SD\;\left(T2-T0\right)}\;$$[mean postoperative score (**T2**)—mean preoperative score (**T0**)/standard deviation (**SD**) of the change in score].

The effect size (**ES**) was calculated with the formula:(2)$$ES=\frac{Mean\;T2-Mean\;T0}{SD\;T0}\;$$[mean postoperative score (**T2**)—mean preoperative score (**T0**)/standard deviation (**SD**) of preoperative score]. 

According to the literature, we considered a small effect 0.2–0.49, moderate effect 0.5–0.79, and large effect ≥ 0.8 [29,30].

### 2.4. Statistical Analysis

The sample size was calculated using the ICC value [31]. Assuming a one-sided error α = 0.05 and a power of 80% (β = 0.20), a null hypothesis ρ0 = 0.3 and an alternative hypothesis ρ1 = 0.7, a population of 22 patients for each group would be required for validation. This sample size was consistent with prior patient-reported validation and adaptation studies [17,18,21,24,32].

Continuous variables were represented by means and standard deviation (SD), whilst categorical and ordinal data were expressed by absolute value and percentage. The mean difference in psychometric properties of PODCI between adolescent self-report and parent-report forms was examined using linear regression models controlling the standard errors with the clustered sandwich estimator to account for repeated measures on the same subject [33].

All analyses were performed with SPSS v. 22.0 (SPSS, Chicago, IL, USA) and STATA v. 11.2 (STATA Corp., College Station, TX, USA). 

## 3. Results

### 3.1. Translation and Cross-Cultural Adaptation

The two final versions of the PODCI are reported in Appendix A.

The translations, checking, and pilot-testing processes of the questionnaires raised minor discrepancies or understanding problems, that were resolved by consensus. In question 2, we changed “half-gallon” to “un litro” since one-liter containers are more commonly available in Italy, although the weight of this carton is different from the original half-gallon container. Similarly, in question 22, we changed “mile” to “km”, since the metric system is more widely used and understandable in Italy, although 1 mile is longer than 1 km.

### 3.2. Demographic and Clinical Characteristics

A total of 59 patients (30 males and 29 females) were involved in the study and completed the questionnaire at T0, T1 and T2 assessment. The mean age was 11.5 ± 2.6 years (range 2–17 years). Thirty-nine participants (66.1%) were adolescents, while twenty (33.9%) were children. The average BMI was 20.8 kg/m^2^.

Flexible flat foot was the most frequent diagnosis subtype (30, 50.8%), and the main surgical procedure was subtalar arthroeresis (30, 50.8%). The most involved site was ankle/foot (34, 57.6%) with a prevalence of bilateral impairments (40, 67.8%). A comprehensive description of participants’ demographics and clinical characteristics is reported in Table 1.

Overall, sex and diagnosis were not significantly associated with any domain, while age correlated with UE and E domains at T0 (Appendix A).

### 3.3. Psychometric Properties

The psychometric properties of adolescent self-report and pediatric (parent-report) PODCI questionnaires are listed in Table 2 and Table 3.

In general, the two versions exhibited an acceptable internal consistency (Cronbach’s α = 0.60–0.92).

The test-retest reliability varied from moderate to acceptable reproducibility in most questionnaire domains. In particular, the ICC fluctuated between 0.31 and 0.89 for self-report and from 0.49 to 0.87 for pediatric. It was low for the E and PC subscales for adolescents, while, in the parent-report form, H and PC domains showed the lowest ICC. Similarly, the SEM varied from 2.92 to 20.80 and from 3.96 to 17.79 for adolescent and parent form, respectively, with the larger spread in E and PC subscales in the adolescent questionnaire, and PC and H for the pediatric version.

No floor effects were seen in the two final versions of Italian PODCI. However, significant ceiling effect were observed in the UE and TBM subscales, both in the self- and parent-reported version.

Overall, moderate correlation was observed between the HSS Pedi-FABS and SPF/GF domains in the two versions of Italian PODCI. Specifically, Spearman’s ρ varied between 0.51 and 0.31 for the adolescent version and between 0.66 and 0.39 for the pediatric version.

In general, we observed small to moderate responsiveness to change (SRM = 0.02–0.48; ES = 0.03–0.34). No significant differences were reported between self- and parent-report responses across all the domains (Table 4).

## 4. Discussion

In the present study, the Italian version of PODCI for adolescent self-report and parent-report was developed, and the cross-cultural adaptation was performed according to international guidelines [22,23].

Internal consistency and test-retest reliability were acceptable for all domains. Overall, these findings were comparable to those observed in the original version and other linguistic adaptations of the PODCI [4,18,19,20,21] and confirmed its suitability in children and adolescents with musculoskeletal disorders (Table 5a,b) [5,6,7,8,9,10,11,12,13,14,15].

Nevertheless, some domains exhibited only moderate test-retest reliability. In particular, the PC subscale showed low values of ICC and high values of SEM and SDC both in the pediatric and adolescent versions, consistently with previous studies [18,19,20,21]. Kwon et al., in the Korean language adaptation, assumed that this could be due to both with the construct of the questionnaire itself, with some item that could affect the psychometric properties of the test and with the subjective perspective of this scale [18]. Another explanation could be given by the choice of administering the T1 session too close to the hospital admission; this could have affected the patient’s anxiety of upcoming surgery. However, poor to moderate reliability was also reported in other adaptation studies [19,20,21], highlighting the need for additional research focused on this aspect.

We found a significant ceiling score in UE and TBM domains. This finding likely reflects the reason for hospital admission in our study, since most children in our cohort, had conditions such as congenital flatfoot or genu valgum that do not interfere with the upper limbs function and the basic mobility. Floor and ceiling effects were explicitly reported by other studies [20,34]. van der Holst and colleagues investigated 10 children undergoing surgery for neonatal brachial plexus palsy. They did not report relevant floor effects for any subscale but observed significant ceiling effect for TBM, SPF, PC and H [20]. In their study on children with spastic cerebral palsy, McCarthy and colleagues found 47% ceiling effect in the PC domain [34]. All these findings confirmed that the ceiling effect of each subscale may be influenced by the specific disease of the cohort investigated.

In terms of criterion validity, the SPF and GF domains showed moderate correlation with the Italian version of the HSS Pedi-FABS. This result was expected, since HSS Pedi-FABS is specifically designed for assessing sports and physical activity in children. Construct and criterion validity were investigated also by other authors for other skeletal disorders. In the Dutch translation [16], van der Holst and coll. anchored the PODCI to clinical tests such as active range of motion, Mallet score and Assisting Hand Assessment. They reported moderate to excellent construct validity for UE, PC, H and GF. In addition, Bae et al. found similar results, investigating children with brachial plexus palsy [7], while McCarthy and colleagues found good to excellent correlation between TBM and UE subscales and GMFM score for cerebral palsy [34]. All these studies highlighted that PODCI is a tool responsive to the predominant clinical condition.

In our study, all subscales of the PODCI showed poor responsiveness to change. This could be explained by the time interval between T0 and T2 (averagely 3 months) that is insufficient for experiencing a significant clinical improvement following the surgical procedures investigated (mostly growth modulation by hemiepiphyseal stapling or subtalar arthroeresis). However, scarce responsiveness of PODCI to surgical treatment was also reported by other authors investigating surgery for cerebral palsy [20,35]. In this perspective, further studies with widest and most homogeneous cohort of patients followed for longer periods are ongoing in our hospital to assess validity and responsiveness of this questionnaire in children with a variety of musculoskeletal disorders, including lower limb deformities.

This study has some limitations. Firstly, the study was not designed as a validation study for a specific orthopedic condition. Although this approach is consistent with Daltroy and colleagues’ original study [4] and some of the further cross-cultural adaptation and validation studies [18,21], additional research is needed to validate the PODCI in selected cohorts of patients with specific conditions. Secondly, in our cohort, all patients presented lower limb anomalies (mostly knee and/or foot and ankle deformities), this limits the considerations regarding the psychometric properties of the Upper Limb and Basic Mobility subscales. Finally, despite the sample size was estimated for testing the ICC, this number could be small to assess the other properties of the PODCI.

## 5. Conclusions

In conclusion, the Italian versions of PODCI is a reliable tool for patient assessment, with acceptable psychometric properties that can be used in pediatric population in various settings, from clinical practice to research studies. 

Nowadays PROMs have become widespread in medicine research, allowing the evaluation of specific pathologies or body regions. Nevertheless, current evidence suggests not using adult PROMs in pediatrics population cause their lack of validity. 

We encourage further research for testing validity and responsiveness of PODCI in children with various musculoskeletal disorders, or creating PROMs suitable for the pediatric population, in order to make the results of future studies more homogeneous.

## Figures and Tables

**Table 1 children-09-00853-t001:** Patient demographics and clinical characteristics.

Characteristic		Overall Population (*n* = 59)
Sex, *n*. (%)		
Male		30 (50.8%)
Female		29 (49.2%)
Age category, *n*. (%)		
Children (age 2–10), *n*. (%)		20 (33.9%)
Adolescent (age 11–18), *n*. (%)		39 (66.1%)
Age, year	Mean ± SD ^¶^ (range)	11.5 ± 2.6 (2–17)
BMI ^⸙^, kg/m^2^	Mean ± SD (range)	20.8 ± 4.4 (13.6–34)
Diagnosis, *n*. (%)		
Flexible flat foot		30 (50.8%)
Genu valgum		12 (20.3%)
Multiple osteochondromas		5 (8.5%)
Clubfoot recurrence		2 (3.4%)
Discoid meniscus		2 (3.4%)
Leg length discrepancy		2 (3.4%)
ACL ^a^ failure		1 (1.7%)
Calcaneal bone cyst		1 (1.7%)
Osteoid osteoma		1 (1.7%)
Legg-Calvé-Perthes disease		1 (1.7%)
Patellar instability		1 (1.7%)
Talocalcaneal coalition		1 (1.7%)
Involved site, *n*. (%)		
Hip		1 (1.7%)
Knee		24 (40.7%)
Ankle/foot		34 (57.6%)
Laterality, *n*. (%)		
Bilateral		40 (67.8%)
Left		9 (15.3%)
Right		10 (16.9%)
Surgical treatment, *n*. (%)		
Subtalar arthroeresis		31 (52.5%)
Temporary epiphyseal/hemiepiphyseal stapling		14 (23.7%)
Osteotomy		7 (11.9%)
Meniscal Saucerization		2 (3.4%)
ACL reconstruction		1 (1.7%)
Bone grafting		1 (1.7%)
Patellar realignment		1 (1.7%)
Radiofrequency ablation		1 (1.7%)
Tendon transfer		1 (1.7%)

^¶^ SD: Standard Deviation; ^⸙^ BMI: Body Mass Index; ^a^ ACL: Anterior Cruciate Ligament.

**Table 2 children-09-00853-t002:** Psychometric properties of Pediatric Outcome Data Collection Instrument (PODCI) Adolescent self-reported form (*n* = 39).

PODCI Subscale	Cronbach’s α *	T0 ^¶^ vs. T1 ^⸙^ ICC ^a^	T0 vs. T1 SEM ^b^	T0 vs. T1 SDC ^c^	FE *^d^	CE *^e^	T0 vs. T2 ^‡^ SRM ^f^	ES ^g^	Correlation with HSS Pedi-FABS *^h^
Upper Extremity Function	0.73	0.85	2.92	8.09	0.0%	51.3%	0.48	0.34	-
Transfers & Basic Mobility	0.68	0.79	3.34	9.25	0.0%	48.7%	0.14	0.15	-
Sport/Physical Functioning	0.78	0.75	9.89	27.43	0.0%	0.0%	0.25	0.25	0.51
Pain/Comfort	0.89	0.45	18.09	50.15	0.0%	20.5%	0.26	0.27	-
Happiness	0.77	0.79	6.65	18.43	0.0%	23.1%	0.02	0.03	-
Expectations	0.90	0.31	20.80	57.66	0.0%	30.8%	0.28	0.25	-
Global Functioning	0.90 ^¥^	0.89	4.23	11.72	0.0%	0.0%	0.04	0.04	0.38

***** T0 measurement; ^¶^ T0: baseline measurement; ^⸙^ T1: 7-days follow-up measurement; ^‡^ T2: 3-month surgical treatment follow-up measurement; ^a^ ICC: Intraclass coefficient; ^b^ SEM: Standardized error of measurement; ^c^ SDC: Smallest detectable change; ^d^ FE: Floor effect; ^e^ CE: Ceiling effect; ^f^ SRM: Standardized response mean; ^g^ ES: Effect size; ^h^ HSS Pedi-FABS: Hospital for Special Surgery Pediatric Functional Activity Brief Scale; ^¥^ Based on 30 patients.

**Table 3 children-09-00853-t003:** Psychometric properties of Pediatric Outcome Data Collection Instrument (PODCI) Parent-reported form (*n* = 20).

PODCI Subscale	Cronbach’s α *	T0 ^¶^ vs. T1 ^⸙^ ICC ^a^	T0 vs. T1 SEM ^b^	T0 vs. T1 SDC ^c^	FE *^d^	CE *^e^	T0 vs. T2 ^‡^ SRM ^f^	ES ^g^	Correlation with HSS Pedi-FABS *^h^
Upper Extremity Function	0.60	0.65	4.90	13.59	0.0%	50.0%	0.07	0.11	-
Transfers & Basic Mobility	0.89	0.87	3.96	10.96	0.0%	55.0%	0.07	0.05	-
Sport/Physical Functioning	0.81	0.68	11.59	32.11	0.0%	0.0%	0.39	0.40	0.66
Pain/Comfort	0.92	0.54	17.79	49.30	0.0%	20.0%	0.28	0.26	-
Happiness	0.73	0.49	12.65	35.07	0.0%	10.0%	0.17	0.18	-
Expectations	0.84	0.77	7.03	19.48	0.0%	30.0%	0.04	0.04	-
Global Functioning	0.90	0.81	5.63	15.60	0.0%	0.0%	0.04	0.03	0.39

***** T0 measurement; ^¶^ T0: baseline measurement; ^⸙^ T1: 7-days follow-up measurement; ^‡^ T2: 3-month surgical treatment follow-up measurement; ^a^ ICC: Intraclass coefficient; ^b^ SEM: Standardized error of measurement; ^c^ SDC: Smallest detectable change; ^d^ FE: Floor effect; ^e^ CE: Ceiling effect; ^f^ SRM: Standardized response mean; ^g^ ES: Effect size; ^h^ HSS Pedi-FABS: Hospital for Special Surgery Pediatric Functional Activity Brief Scale.

**Table 4 children-09-00853-t004:** Mean difference on psychometric properties of Pediatric Outcome Data Collection Instrument (PODCI) between Adolescent self-report and Parent-report forms. Repeated measures linear regression models.

PODCI Subscale	MD ^⸙^	95%CI ^¶^	*p*-Value
Upper Extremity Function.	−3.51	−9.15; 2.13	0.218
Transfers & Basic Mobility	−0.96	−5.99; 4.07	0.703
Sport/Physical Functioning	0.94	−9.38; 11.25	0.856
Pain/Comfort	2.22	−6.41; 10.84	0.609
Happiness	−2.55	−10; 4.91	0.497
Expectations	4.46	−3.86; 12.78	0.288
Global Functioning	−0.22	−6.32; 5.87	0.942

^⸙^ MD: Mean difference; ^¶^ CI: Confidence interval.

**Table 5 children-09-00853-t005:** (**a**) Comparison between the Original English, Korean, Brazilian, Dutch, Turkish, and Italian Pediatric Outcome Data Collection Instrument (PODCI) on population and criterion validity. (**b**) Subscales’ comparison between the Original English, Korean, Brazilian, Dutch, Turkish, and Italian Pediatric Outcome Data Collection Instrument (PODCI) on reliability.

**(a)**							
**Psychometric Property**		**English PODCI [2]**	**Korean PODCI [16]**	**Brazilian PODCI [17]**	**Dutch PODCI [18]**	**Turkish PODCI [19]**	**Italian PODCI**
Population	Age, year	2–18	5–18	2–18	2–10	2–18	2–17
	Participants	Adolescent Parent	Adolescent Parent	Adolescent Parent	Parent	Parent	Adolescent Parent
	Diagnosis	All orthopedic disorders	Affected by Cerebral palsy vs. Healthy	Affected by Juvenile idiopathic arthritis	Affected by Neonatal brachial plexus palsy	Common orthopedic conditions	All orthopedic conditions required surgical treatment
Criterion validity		Good correlation with CHQ PF-28 ^¶^ physical function	-	Good correlation with CHQ PF-28 physical function	Moderate to strongly correlation with aROM ^⸙^ Mallet score ^‡^ AHA ^†^	-	Moderate correlation with HSS Pedi-FABS ^¥^
**(b)**							
**Subscale**	**Reliability**	**English PODCI [2]**	**Korean PODCI [16]**	**Brazilian PODCI [17]**	**Dutch PODCI [18]**	**Turkish PODCI [19]**	**Italian PODCI**
**Adolescent self-reported form**							
Upper Extremity Function	Internal consistency, α *	0.84	0.264	NR	-	-	0.73
	Test-retest reliability, ICC ^⸙^	0.96	NR	0.97	-	-	0.85
Transfers & Basic Mobility	Internal consistency, α	0.91	0.824	NR	-	-	0.68
	Test-retest reliability, ICC	0.97	NR	0.81	-	-	0.79
Sport/Physical Functioning	Internal consistency, α	0.9	0.891	NR	-	-	0.78
	Test-retest reliability, ICC	0.87	NR	0.97	-	-	0.75
Pain/Comfort	Internal consistency, α	0.84	0.075	NR	-	-	0.89
	Test-retest reliability, ICC	0.89	NR	0.59	-	-	0.45
Happiness	Internal consistency, α	0.76	0.877	NR	-	-	0.77
	Test-retest reliability, ICC	0.87	NR	0.53	-	-	0.79
Expectations	Internal consistency, α	0.9	-	-	-	-	0.9
	Test-retest reliability, ICC	0.76	-	-	-	-	0.31
Global Functioning	Internal consistency, α	0.92	0.802	0.82	-	-	0.9
	Test-retest reliability, ICC	0.95	NR	NR	-	-	0.89
**Parent-reported form (for pediatric and adolescent)**							
Upper Extremity Function	Internal consistency, α	0.94	0.762 ^a^/0.912 ^b^	NR	0.695	0.9	0.6
	Test-retest reliability, ICC	0.94	NR	0.89 ^c^/0.86 ^d^	0.972	0.94	0.65
Transfers & Basic Mobility	Internal consistency, α	0.95	0.740 ^a^/0.967 ^b^	NR	0.667	0.95	0.9
	Test-retest reliability, ICC	0.96	NR	0.97 ^c^/0.04 ^d^	0.636	0.97	0.87
Sport/Physical Functioning	Internal consistency, α	0.93	0.855 ^a^/0.918 ^b^	NR	0.597	0.93	0.81
	Test-retest reliability, ICC	0.93	NR	0.97 ^c^/0.60 ^d^	0.948	0.95	0.68
Pain/Comfort	Internal consistency, α	0.87	0.545 ^a^/0.371 ^b^	NR	0.161	−0.38	0.92
	Test-retest reliability, ICC	0.83	NR	0.82 ^c^/018 ^d^	0.022	0.07	0.54
Happiness	Internal consistency, α	0.82	0.864 ^a^ /0.793 ^b^	NR	0.928	0.77	0.73
	Test-retest reliability, ICC	0.71	NR	0.69 ^c^/0.68 ^d^	0.96	0.82	0.49
Expectations	Internal consistency, α	0.88	-	-	-	-	0.84
	Test-retest reliability, ICC	0.83	-	-	-	-	0.77
Global Functioning	Internal consistency, α	0.94	0.858 ^a^/0.968 ^b^	0.90 ^c^/0.72 ^d^	0.781	0.96	0.9
	Test-retest reliability, ICC	0.97	NR	NR	0.803	0.98	0.81

^¶^ CHQ PF-28: Short Child Health Questionnaire Parent Form-28 items; ^⸙^ aROM: active Range of Motion; ^‡^ Mallet score: functional scoring system to assess shoulder abduction/external rotation deficits in children with obstetric brachial plexus palsy; ^†^ AHA: Assisting Hand Assessment; ^¥^ HSS Pedi-FABS: Pediatric Functional Activity Brief Scale. * α: Cronbach’s α; ^⸙^ ICC: Intraclass coefficient; NR: Not reported; ^a^ Pediatric and Adolescent (parent-reported) forms healthy parents; ^b^ Pediatric form cerebral palsy parents; ^c^ Pediatric form; ^d^ Adolescent (parent-reported) form.

## Data Availability

The datasets generated during and/or analysed during the current study are available from the corresponding author on reasonable request.

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
