# Peer review of "Translation and Cross-Cultural Adaptation of the Pediatric Outcomes Data Collection Instrument into the Italian Language"

_children, 2022, doi:10.3390/children9060853_

Round 1
Reviewer 1 Report
The presented piece of a research is a standard cultural adapptation, with significant issues
The sample size is small with wide ranges of age and medical conditions- some of them has short and some of them long convalescence.
Due to material weakness, parameters such as Cronbach or ICC are just at the level of acceptance.
Author Response
Point 1: The presented piece of a research is a standard cultural adaptation, with significant issues.
The sample size is small with wide ranges of age and medical conditions- some of them has short and some of them long convalescence.
Due to material weakness, parameters such as Cronbach or ICC are just at the level of acceptance.
Response 1: You brought up a good point. Our small sample size could be a methodological constraint of our study, as we did not test the convergent validity. Nonetheless, the present study is a cross-cultural adaptation study, the sample size is comparable to other PODCI cross-cultural adaptation studies, such as the Brazilian study, which included 57 subjects (32 children and 25 adolescents) (do Monte et al., BMC Pediatrics 2013); additionally, our patient sample may be considered representative of the general population that is normally referred to our Unit of Pediatric Orthopedics and Traumatology.
Reviewer 2 Report
It’s an interesting and original paper
Author Response
Point 1: It’s an interesting and original paper
Response 1: On behalf of all co-authors, thank you for your feedback.
Reviewer 3 Report
I found error in line 173-174 - You wrote that group of children consist of 39 patient and adolescent - 20. In Table 1 the data are opposite.
The paper is well written. Congratulation. It's a pity, that refers only to patient after lower limb procedures.
Author Response
Point 1: I found error in line 173-174 - You wrote that group of children consist of 39 patient and adolescent - 20. In Table 1 the data are opposite.
Response 1: Thanks to your advice, we fixed the typo in the manuscript.
Point 2: The paper is well written. Congratulation. It's a pity, that refers only to patient after lower limb procedures.
Response 2: On behalf of all co-authors, thank you for your feedback. We both agree that the lack of patients with upper limb involvement is regrettable. Unfortunately, during the enrollment period, we only treated patients with lower limb impairments who required surgical intervention. Despite this, previous research, most notably Wall et al. (2020) and Tanrverdi et al. (2021), validated the questionnaire for upper limb deficits.
Reviewer 4 Report
This manuscript is aimed to translate, cross-culturally adapt, and assess the psychometric properties of the PODCI score in the Italian pediatric population. The PODCI questionnaire was culturally adapted to Italian patients in accordance with the literature guidelines. The study included 59 participants from single orthopedic institution, who underwent orthopedic surgery for various skeletal conditions. The questionnaire was administered to participants in multiple time-point (T0, T1, T2). Internal consistency was evaluated using Cronbach's alpha. Reproducibility was assessed using the intraclass correlation coefficient (ICC) between T0 and T1 assessment. Criterion validity was assessed using Spearman’s correlation coefficients between PODCI and the Hospital for Special Surgery Pediatric Functional Activity Brief Scale (HSS Pedi-FABS). Responsiveness was evaluated by difference between T0 and T2 using the effect size (ES) and the standardized response mean (SRM) calculation.
I read the article with interest, the title is well thought out and faithfully reflects the content of the study, but the characteristics of the study should be specified. The abstract is adequately developed, and it is useful to frame the purpose of the study.
In the introduction, the Patient-Reported Outcomes Measurements (PROMs) have been shortly described. In the material and methods have been adequately developed. The discussion is sufficiently developed.
Nevertheless, some minor changes are needed to be considered suitable for publication.
Comment 1: The characteristics of the study should be specified.
Comment 2: In the introduction: " The PODCI is made up of three different formats: filled by parents for children, filled by parents for adolescents, and self-report for adolescents. The questionnaire is multidimensional, consisting of 86 questions (83 questions in adolescent self-report) that investigate the following domains: ‘upper extremity and physical function’ (UE), ‘transfer and basic mobility’ (TBM), ‘sports and physical function’ (SPF), ‘pain/comfort’ (PC), ‘happiness’ (H), ‘global functioning’ (GF) and ’expectations from treatment and satisfaction’ (E). Mean score for each domain is reported on a 0 (worst result) to 100 (best result) scale.". It may be useful to include some bibliographical references.
Comment 3: In the introduction: I don’t understand if in your study you talk only about congenital disorders or refer also to fractures in pediatric age, I would clarify this aspect. Eventually it is necessary to add bibliographic references. For Example: (Pavone V. et al (2020) "Analysis of loss of reduction as risk factor for additional secondary displacement in children with displaced distal radius fractures treated conservatively") or (Vescio A, et al. (2022) "Is Obesity a Risk Factor for Loss of Reduction in Children with Distal Radius Fractures Treated Conservatively?").
Comment 4: In materials and methods: It would be appropriate to refer to the criteria of inclusion and exclusion of patients in the study, they do not seem to be very clear.
Comment 5: The limits of the study are clear enough. However, I would add a brief description of what could be improved in future studies on the same topic.
Comment 6: Finally, additional English editing is needed. The Non-Native Speakers of English Editing Certificate was not signed.
Author Response
Point 1: The characteristics of the study should be specified.
Response 1: We upgraded the study’ characteristics in the Methods paragraph of the manuscript.
Point 2: In the introduction: " The PODCI is made up of three different formats: filled by parents for children, filled by parents for adolescents, and self-report for adolescents. The questionnaire is multidimensional, consisting of 86 questions (83 questions in adolescent self-report) that investigate the following domains: ‘upper extremity and physical function’ (UE), ‘transfer and basic mobility’ (TBM), ‘sports and physical function’ (SPF), ‘pain/comfort’ (PC), ‘happiness’ (H), ‘global functioning’ (GF) and ’expectations from treatment and satisfaction’ (E). Mean score for each domain is reported on a 0 (worst result) to 100 (best result) scale.". It may be useful to include some bibliographical references.
Response 2: In response to your suggestion, we included bibliographical references to the work (line 64).
Point 3: In the introduction: I don’t understand if in your study you talk only about congenital disorders or refer also to fractures in pediatric age, I would clarify this aspect. Eventually it is necessary to add bibliographic references. For Example: (Pavone V. et al (2020) "Analysis of loss of reduction as risk factor for additional secondary displacement in children with displaced distal radius fractures treated conservatively") or (Vescio A, et al. (2022) "Is Obesity a Risk Factor for Loss of Reduction in Children with Distal Radius Fractures Treated Conservatively?").
Response 3: In the introduction, we clarified this point.
Point 4: In materials and methods: It would be appropriate to refer to the criteria of inclusion and exclusion of patients in the study, they do not seem to be very clear.
Response 4: To clarify this issue, we have added two subheadings to the Study Design paragraph: Participants (lines 108-113) and Survey design (lines 114-125).
Point 5: The limits of the study are clear enough. However, I would add a brief description of what could be improved in future studies on the same topic.
Response 5: We incorporated your recommendations and revised the content (lines 295-301).
Point 6: Finally, additional English editing is needed. The Non-Native Speakers of English Editing Certificate was not signed.
Response 6: Thank you for your advice. We made extensive linguistic revisions.
Reviewer 5 Report
Important, well done.
Author Response
Point 1: Important, well done.
Response 1: On behalf of all co-authors, thank you for your feedback.
Round 2
Reviewer 1 Report
In my opinion, only English editing was done. The analyzed material- participants are still incoherent- of various ages and surgical treatment methods. Thus the main issue -drawback needs to be changed.
Author Response
Point 1: In my opinion, only English editing was done. The analyzed material- participants are still incoherent- of various ages and surgical treatment methods. Thus the main issue -drawback needs to be changed.
Response 1: On behalf of the all authors, thank you for your feedback.
Patient-reported outcome measures (PROMs) are the most advanced instruments for monitoring patients’ health status, with higher reliability, validity, and sensitivity to symptom change than most legacy health measures.
Many PROMs for adults have been developed and translated into several languages, including Italian. However, no formal translated questionnaire into Italian that addresses pediatric patients with orthopedic diseases on a multidimensional level is currently available. This is a critical point for patients with several related disorders for whom uni-dimensional evaluation may be insufficient for a holistic health assessment.
Therefore, the RAMS (Rete Apparato Muscolo-Scheletrico), an Italian network comprised of the main leading national research institutions in pediatric orthopedics, launched an initiative to develop standardized new patient-reported measures for national use in children with musculoskeletal disorders. The ERN-BOND, the European Reference Network on Rare Bone Diseases, has likewise supported this initiative.
The main purpose of the present study was to provide an official translation and cross-cultural adaptation in Italian of the PODCI that could be used on digital platforms and electronic devices to share data across the network's hospitals.
So far, the study has not been designed as a validation study for a specific orthopedic condition. This is consistent with Daltroy and colleagues' original study (Daltroy et al., 1998), in which the PODCI questionnaire was administered to a pediatric population with various orthopedic conditions. Other studies that reported translations of several pediatric PROMs (PEDI-CAT, APPT,...) were performed in similar cohorts with different disorders.
However, this is a pilot study. Our research is ongoing to validate the PODCI in selected cohorts of patients with specific conditions and compare the PODCI with other pediatric PROMs.
We revised the manuscript to clarify our intentions.